



# Assessing the accuracy of low-cost optical particle
# sensors using a physics-based approach
David H Hagan[1, 2], Jesse H Kroll[1, 3]
[1]Department of Civil and Environmental Engineering, Massachusetts Institute of Technology,
Cambridge, MA 02139, USA
[2]QuantAQ, Inc., Somerville, MA 02143, USA
[3]Department of Chemical Engineering, Massachusetts Institute of Technology, Cambridge, MA
02139, USA
Corresponding Author Emails: david.hagan@quant-aq.com or jhkroll@mit.edu



# 1    Abstract

Low-cost sensors for measuring particulate matter (PM) offer the ability to understand
human exposure to air pollution at spatiotemporal scales that have previously been
impractical. However, such low-cost PM sensors tend to be poorly characterized, and
their measurements of mass concentration can be subject to considerable error. Recent
studies have investigated how individual factors can contribute to this error, but these
studies are largely based on empirical comparisons and generally do not examine the
role of multiple factors simultaneously. Here, we present a new physics-based framework
and open-source software package (*opcsim*) for evaluating the ability of low-cost optical
particle sensors (optical particle counters and nephelometers) to accurately characterize
the size distribution and/or mass loading of aerosol particles. This framework, which uses
Mie Theory to calculate the response of a given sensor to a given particle population, is
used to estimate the relative error in mass loading for different sensor types, given
variations in relative humidity, aerosol optical properties, and the underlying particle size
distribution. Results indicate that such error, which can be substantial, is dependent on
the sensor technology (nephelometer vs. optical particle counter), the specific
parameters of the individual sensor, and differences between the aerosol used to
calibrate the sensor and the aerosol being measured. We conclude with a summary of
likely sources of error for different sensor types, environmental conditions, and particle
classes, and offer general recommendations for choice of calibrant under different
measurement scenarios.

# 23    1.    Introduction

Human exposure to aerosols is associated with adverse health impacts and increased
mortality (Apte et al., 2018; Burnett et al., 2018; Cohen et al., 2017; Dockery et al., 1993).
The source and composition of aerosols has been linked to a range of negative health



impacts (Antonini et al., 2003; Hart et al., 2012; Henneberger and Attfield, 1997; Lipsett
and Campleman, 1999), with more than 4 million annual deaths worldwide attributed to
ambient particulate matter pollution (Cohen et al., 2017). Accurate estimates of aerosol
sources and health impacts rely critically on measurements of particulate matter
concentrations across indoor and outdoor environments worldwide.
In many countries, particulate matter (PM) pollution is regulated by national or local
government agencies (e.g., the US EPA in the United States) and is typically measured
using federally-approved reference methods that are high in accuracy and precision. The
existing infrastructure is generally designed to measure regional-scale air pollution, in
order to enforce (and assess the effectiveness of) air quality regulations. However, particle
pollution can vary in space and time at much finer resolution than can be measured using
standard monitoring technologies, given their relatively high cost and size. Over the past
several years, new technologies have emerged at price points (<\$2000) that allow PM
measurements to be made with much higher spatiotemporal resolution, even down to
the individual human level (Koehler et al., 2019; Tryner et al., 2019a, 2019b). These devices
are physically small, use very little power, and can easily be deployed at scale. As a result,
such sensors are ideally suited for use in dense distributed sensor networks, providing
high-resolution air quality measurements, as well in as personal monitoring, providing
individuals with the ability to measure and understand their exposure to harmful air
pollutants. As with all low-cost sensors (LCS), accuracy is of paramount concern; as shown
by a number of recent laboratory and field-based evaluation studies (Crilley et al., 2018;
Dacunto et al., 2015; Di Antonio et al., 2018; Holstius et al., 2014; Levy Zamora et al., 2019;
Malings et al., 2020; Northcross et al., 2013; Sousan et al., 2016b, 2016a; Wang et al.,
2015), PM sensors can perform quite poorly without additional constraints or calibrations.





Most low-cost PM sensors measure particles via light scattering. Sampled particles
intercept a beam of light (typically from a laser or LED with a wavelength between 405
and 780 nm), and the scattered light is measured and correlated to a PM concentration.
In this work, we refer to such instruments as Optical Particle Sensors (OPS's). OPS's can
be broken down into two main types, nephelometers and Optical Particle Counters
(OPC's). Nephelometers measure the particles as an ensemble, gathering light scattered
by all particles across a wide range of angles, typically 7°-173° to avoid pure forward and
backward scattering (Abu-Rahmah et al., 2006; Ahlquist and Charlson, 1967; Anderson et
al., 1996). The total scattering amplitude is then correlated to a mass measurement made
by a reference instrument. (Nephelometers that measure scattered light at a single angle
are sometimes referred to as photometers; for the purposes of this work we consider
photometers to be a subclass of nephelometer.) OPC's, by contrast, detect particles
individually, providing information on their number and size. Light scattered by each
individual particle is measured and each pulse is assigned to a size bin based on its total
light intensity, resulting in a histogram which is converted to a mass loading once the
entire distribution has been measured. While these technologies have been around for
decades (Gucker et al., 1947; Patterson et al., 1926), they have recently become available
at much lower cost due to the availability of small, inexpensive light sources and
electronic components.
The use of light scattering introduces a number of fundamental limitations for making PM
mass measurements. Many of these arise from environmental conditions and/or the
properties of the aerosol being measured; these can be especially problematic when
calibration is done using only a single aerosol type or condition. A number of recent
empirical studies of OPS's have investigated some of these limitations. These issues
include: (1) the inability to adapt to changes in the particle size distribution (Dacunto et





al., 2015; Wang et al., 2015); (2) the hygroscopic growth of particles due to changes in
ambient relative humidity (Crilley et al., 2018; Di Antonio et al., 2018; Malings et al., 2020;
Zheng et al., 2018); (3) changes in scattering efficiency due to changes in aerosol optical
properties (Crilley et al., 2018; Di Antonio et al., 2018); and (4) the need for aerosol-specific
correction factors to account for changes in density (Dacunto et al., 2015; Northcross et
al., 2013). While these studies have examined how these individual effects in isolation may
affect PM accuracy, to our knowledge there has not been a systematic, comprehensive
investigation of all these factors in total. Complicating matters is the fact that these
individual properties are all intertwined – for example, when relative humidity increases,
it can cause particles to take up water, which can change not only their size and mass but
also their shape, refractive index, and density.
To disentangle the relative contribution of error by various interacting sources, we have
developed a model that describes how a given sensor will respond to different aerosols
under most conditions. This model is based entirely on the underlying physics of light
scattering (Mie Theory) rather than empirical relationships obtained through laboratory
or field measurements. While previous work has modeled nephelometers and OPC's in a
similar way (Walser et al., 2017), we believe this is the first detailed treatment of light
scattering as it relates specifically to LCS. We use this model to isolate the relevant
sources of error and develop a better understanding of the limitations (as well as
strengths) of different kinds of OPS's.
The modeling tool described here, which is open source and freely available, can be used
for the systematic study of how different OPS's may detect various aerosol types under a
range of environmental conditions. This enables new insights into the potential errors
associated with a given PM measurement, optimal strategies for calibrating OPS's, and





ultimately in the design of the sensors themselves and the development of algorithms for
data analysis.

## 2.  Methods

The modeling framework described in this section is available as an open-source (MIT
license) python library (*opcsim*) and has been made available on GitHub. Detailed
documentation, including installation instructions and examples, are available online
(Hagan and Kroll, 2019). The framework , called "opcsim", consists of two primary
components: the code that models OPS's and implements the Mie Theory algorithms
(Bohren and Huffman, 1983; Sumlin et al., 2018), and the code to build and evaluate
aerosol distributions.
We follow the same general modeling pattern regardless of sensor type. Steps include:
(1) defining the device based on its key physical parameters; (2) calibrating the device to
a specific aerosol type (for OPC's) or aerosol distribution (for nephelometers); and (3)
evaluating each particle in an aerosol population by computing the scattered light signal
using Mie theory and converting that signal to the sensor output based on its calibration.
In the following sections we describe how the aerosol population is described by the
model, followed by how the sensors themselves are treated.

## 2.1    Representing an aerosol distribution

We represent an aerosol distribution as the sum of *n* lognormal modes, where each mode
*i* is defined by its geometric mean particle diameter ($\overline{D}_{pi}$), geometric standard deviation
($\sigma_i$), and number concentration ($N_i$). The aerosol distribution as a function of diameter $D_p$
($dN/dlogD_p$) is given by Equation 1 (Seinfeld and Pandis, 2006):



$$\frac{dN}{dlogD_p} = \sum_{i=1}^{n} \frac{N_i}{\sqrt{2\pi}log\sigma_i} \exp\left(-\frac{(logD_p - log\bar{D}_{pi})^2}{2 log^2 \sigma_i}\right)$$
(Equation 1)

Additionally, we define the composition of the aerosol distribution by defining the
particle density ($\rho_i$), hygroscopic growth factor ($\kappa_i$), and complex refractive index ($m_i$) for
each mode. The role of these additional parameters is discussed in section 3, below.
While more complex representations of the chemical makeup of the aerosol can be
implemented using our modeling framework (i.e., core-shell representation of aerosols,
complex aerosol mixtures, etc.), for the purposes of this manuscript we focus only on well-
mixed homogeneous particle modes, as described by Eq. 1. The above number
distribution can be converted to a mass distribution (or total mass concentration) by
assuming all particles are spherical with a known density (Seinfeld and Pandis, 2006).
## 2.2      Representing Optical Particle Sensors
### 2.2.1    Optical Particle Counters (OPC's)
An OPC is defined by three instrument-specific parameters: (1) the wavelength of the
light source ($\lambda$), (2) the viewing angle for which the scattered light is collected, and (3) the
number of discrete size bins and their widths. A bin, in this context, refers to a single
"slice" of the aerosol size distribution, with a fixed width and units of particle diameter.
Typically, most low-cost OPC's have between 2-30 bins. These can be determined either
by looking up the parameters in the device's datasheet provided by the manufacturer or
by making simple measurements. Bins are often chosen to reduce the uncertainty in
correct bin assignments within the bounds of what the sensor is capable of detecting.
Most low-cost OPCs have the smallest bin at $D_{min}$ ~500 nm, with cost typically being the
driving factor – OPC's with lower $D_{min}$ employ more expensive, higher-quality optics and
photo detectors, allowing them to accurately detect smaller particles. In this work, the



bin boundaries (and hence widths) used for a given OPC are taken from the
manufacturer's spec sheets, if available; otherwise they are calculated by generating an
array of logarithmically-spaced bin boundaries for a set number of bins ($n_{bins}$) between the
minimum and maximum defined diameters ($D_{min}$ and $D_{max}$, respectively). Most often, a
light pulse generated by a single particle is assigned to exactly one bin; however, there
exist approaches where bin assignments are made using a probability distribution (Walser
et al., 2017); this is not implemented in this model but is an approach that could be added
in the future.. Table 1 lists bin widths and other parameters for a few commercially-
available low-cost OPC's.
**Table 1.** Characteristics of a selection of commercially-available low-cost optical particle
counters and nephelometers.

| Manufacturer | OPS Type | Model | $\lambda$ (nm) | Viewing Angle ($\emptyset_1$, $\emptyset_2$) | # of Size Bins |
|---|---|---|---|---|---|
| Alphasense, Ltd. | OPC | OPC-N2 | 658 | (32.0°, 88.0°) | 16 (0.38 – 17.5 µm) |
| Alphasense, Ltd. | OPC | OPC-N3 | 658 | (32.0°, 88.0°) | 24 (0.35 – 40.0 µm) |
| Particle Plus | OPC | | 785 | (58.0°, 118.0°) | 6 (0.3 – 10.0 µm) |
| NOAA/Handix | OPC | POPS | 405 | (38.0°, 142.0°) | 16 (0.132 – 3.65 µm) |
| Plantower | Nephelometer | PMS5003 | ~650 | ?[1] | 6 (0.3 – 10+ µm)[2] |
| Sharp | Nephelometer (Photometer) | GP2Y1010AUOF | 870-980 | ?[1] | 1 (?)[3] |
| Shinyei | Nephelometer (Photometer) | PPD42NS | 870-980 | ?[1] | 1 (>1 µm) |
| Samyoung | Nephelometer (Photometer) | DSM501A | 870-980 | ?[1] | 1 (>1 µm) |

[1] Unknown; not provided in the manufacturer's technical data sheet or the technical literature



[2] The PMS5003 reports six bins; however these are not actual size bins, but rather software-
computed results (He et al., 2020).
[3] No size detection limit for the Sharp sensor is listed in the literature or in the manufacturer's
technical data sheet
OPC's are calibrated by relating the scattered light intensity – a combination of the
particle's scattering cross section ($C_{scat}$) and laser intensity – to the particle diameter.
Practically, this is done by using calibration aerosols with known optical properties and
size and generating a calibration curve between the test aerosol and the electronic pulse
height generated by that aerosol. After repeating this process for many sizes, a calibration
curve can be generated. Here, we compute the $C_{scat}$ values using Mie Theory using
attributes of the calibration aerosol. To simplify the model, we make several assumptions,
including: (1) all particles are spherical and homogeneous (well-mixed); (2) the laser
intensity is constant, implying all particles are perfectly centered in the beam of the laser;
and (3) the photodetector and electronics are 100% efficient, and so we do not consider
the impact of signal-to-noise limitations.
As most low-cost OPC's contain an elliptical re-focusing mirror to gather the scattered
light across many angles, we compute the integrated light scattering intensity following
a procedure first introduced by Jaenicke and Hanusch (Jaenicke and Hanusch, 1993). Mie
theory calculations are implemented using equations by Bohren and Huffman (Bohren
and Huffman, 1983). The scattering cross-section is calculated as:
$$C_{scat} = \frac{\lambda}{4\pi} \int_{\Theta_1}^{\Theta_2} [i_1(\Theta) + i_2(\Theta)] \sin\Theta \, d\Theta \qquad \text{(Equation 2)}$$



where λ is the wavelength of incident light, Θ is the viewing angle (which ranges from $\Theta_1$
to $\Theta_2$), and $i_1$ and $i_2$ are the intensity distribution functions (Bohren and Huffman, 1983).
Figure 1 depicts the calibration curve generated for an OPC with the characteristics of
the Alphasense OPC-N2 (Table 1), using polystyrene latex spheres (PSL's) of different
diameters for calibration. Eq. 2 was used to compute the theoretical $C_{scat}$ values (y axis),
integrated across the entire viewing angle, for a range of particle diameters (x axis). The
$C_{scat}$ values at each bin boundary (green dots in Fig. 1) are then computed, and spline
interpolation is used between each individual bin boundary to generate a mapping
between the scattering amplitude and its corresponding bin assignment. In practice, this
operates as a lookup table – a particle crossing the laser generates a scattering amplitude
which is associated with a specific 'bin' via the calibration.

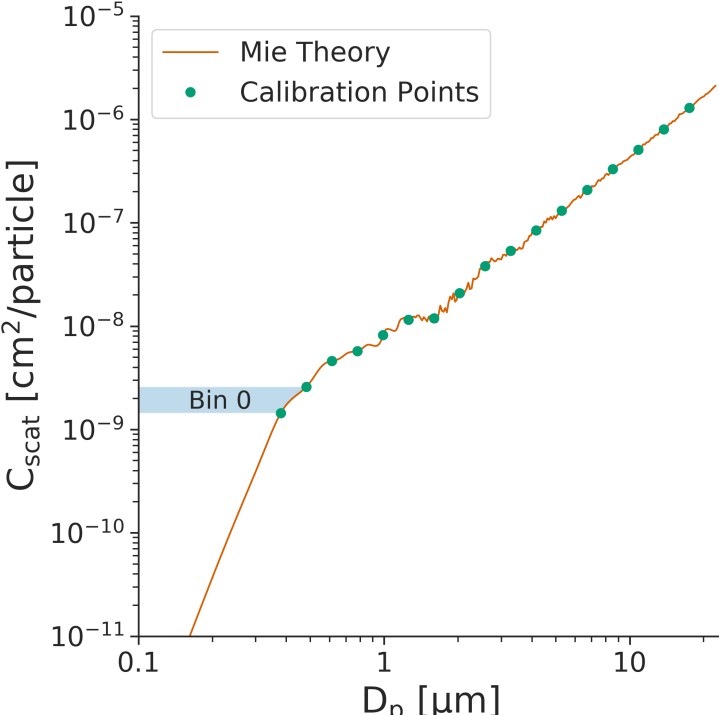

**Figure 1.** Calibration data for an OPC with 16 discrete size bins between 0.38 – 17.5 µm. OPC parameters were chosen to match the Alphasense OPC-N2 (wavelength of 658 nm, viewing angle of 32-88°) using monodispersed polystyrene latex spheres ($m = 1.592 + 0j$). The integrated scattering amplitude calculated using Mie theory is shown as the solid line, with points depicting the corresponding scattering amplitude at each of the bin boundaries. Shown as a shaded box is the range of scattering amplitudes that is assigned to the smallest size bin.

For OPC's that measure scattered light across a wide angle, $C_{scat}$ is generally a
monotonically increasing function of the particle size. However, there may be cases where
this is not true, typically due to the presence of Mie resonance (e.g., near $D_p$=1.5 µm in
Fig. 1). When the function is not monotonic, we apply a smoothing algorithm (Cerni, 1983;
Osborne et al., 2008) or merge together multiple bins (Pinnick et al., 1981; Walser et al.,
2017) and accept the tradeoff where we obtain a higher rate of correct bin assignment in
exchange for reduced bin resolution. This non-monotonicity is less of an issue as the





viewing angle becomes wider, as the larger range of angles will "smooth out" any Mie
resonances (Figure S1). The wide viewing angle thus offers two key advantages: (1) the
total signal (pulse height) is larger, making it easier to detect small particles using
inexpensive electronics; (2) the calibration curve is less susceptible to small changes in
particle scattering cross-section.
While an OPC sizes and counts individual particles, we generally are interested in
evaluating the entire population of particles. To obtain the results for the entire
population, we compute the scattering cross-section for each particle in the distribution,
and assign it to a bin using the calibration curve generated previously – this results in a
histogram with the total sum of particles in each discrete size bin over a period of time.
Once we have the number distribution, we can compute the aerosol mass loading (PM)
using Eq. (3):
$$PM = \rho \sum_i N_i \frac{\pi}{6} d_{p,i}^3 \qquad \text{(Equation 3)}$$
where $N_i$ is the number concentration for a given size bin, $d_{p,i}$ is the geometric mean
diameter for a given size bin, and $\rho$ is the particle density, chosen to be constant. We can
integrate mass loadings between different diameters by summing only across a sub-
selection of bins (for example, if we intend to calculate the $PM_1$ mass concentration, we
would choose only the size bins corresponding to particles sized between 0-1 µm,
whereas to calculate the $PM_{2.5}$ mass concentration, we would use the bins corresponding
to sizes between 0-2.5 µm). This approach for computing mass loadings is similar to that
used by others (Di Antonio et al., 2018), though we use the geometric mean particle
diameter as opposed to the mean particle diameter.



### 2.2.2    Integrating nephelometers

Nephelometers gather the light scattered by an aerosol population across a wide range of angles to gather as much of the scattered light as possible, while avoiding the near-forward and near-backward scattered light. Here, we define a nephelometer by the wavelength of its light source (λ) and its viewing angle.

In practice, nephelometers are calibrated empirically by correlating the total scattered light signal to a reference mass measurement (Dacunto et al., 2015; Sousan et al., 2016b; Wang et al., 2015). Within our model, we do the same by computing the total scattered light signal using Mie theory and then take the ratio of the scattered light to a calculated mass loading. The total scattered light signal is calculated by integrating Eq. 2 across the entire particle size distribution, resulting in a single scattered light intensity for a given aerosol distribution. The calibration factor is then calculated by taking the ratio of this value and the mass loading of the aerosol distribution, which is calculated by integrating the volume distribution and multiplying by the particle density (Equation 3). Once we have computed the calibration factor, we can calculate the mass loading for any aerosol distribution by multiplying the calibration factor by the calculated total scattered light signal.

## 3.    Results and discussion

We use the model described above to isolate the relative source of error associated with various differences in physical and optical properties of aerosols as well as with the devices themselves. We include both simple, targeted experiments probing the effects of changes in isolated properties, as well as more complex, realistic experiments that attempt to mimic real-world scenarios. In the latter case, we include a variety of aerosol types in our model runs to resemble real-world use-cases; aerosol types include urban



aerosol, wildfire emissions, marine aerosol, dust, and continental background. The
physical and optical properties for these aerosols are summarized in Table 2. We discuss
these results in the context of three particle sensors chosen to be representative of low-
cost OPS's: a nephelometer, which uses a 658 nm light source and has a viewing range
of 7°-173°, and two OPC's, both with 16 equally-spaced bins, a 658 nm light source, and
a viewing angle of 32-88°. The two OPC's differ only in the minimum particle size
measured: the 'low-cost OPC' is representative of commercial OPC's currently on the
market and measures particles in the 0.38-17.5 µm size range; and the 'high-end OPC',
representing an idealized OPC that can measure much smaller particles, with a detection
range of 0.1-17.5 µm.
**Table 2**. Aerosol optical and chemical properties used in this work.

| Aerosol Type | Refractive index | Hygroscopicity parameter $\kappa$[6] | Density (g cm$^{-3}$) |
|---:|:---:|:---:|:---:|
| Urban[1] | 1.525+0.020j | 0.40 | 1.35 |
| Background[2] | 1.520+0.008j | 0.25 | 1.45 |
| Marine[3] | 1.384+0.001j | 1.10 | 2.16 |
| Dust[4] | 1.555+0.003j | 0.03 | 2.60 |
| Wildfire[5] | 1.570+0.002j | 0.10 | 1.58 |

[1] (Chen et al., 2019; Cheung et al., 2019; Hussein et al., 2004; Jurányi et al., 2013; Raut
and Chazette, 2007; Rissler et al., 2014; Shepherd et al., 2018; Wehner and
Wiedensohler, 2003)
[2] (Levoni et al., 1997; Wang et al., 2014; Yin et al., 2015)
[3] (Levoni et al., 1997; Ueda et al., 2016; Zieger et al., 2017)
[4] (Koehler et al., 2009; Petzold et al., 2009; Rocha-Lima et al., 2018)
[5] (Bougiatioti et al., 2016; Laing et al., 2016; McMeeking, 2004; Shepherd et al., 2018)
[6] (Petters and Kreidenweis, 2007)



We begin by investigating the impact that water uptake, driven by changes in the ambient
relative humidity, has on the ability of all three OPS's to infer $PM_{2.5}$ mass. Next, we explore
the impact of aerosol optical properties (namely, the complex RI), followed by the impact
that perturbations in the underlying particle size distribution can have in the OPS's ability
to infer mass loadings. Finally, we summarize our results into general recommendations
about each OPS type. Throughout, to provide a simple metric for the accuracy of OPS
measurements, we present our results in terms of the ratio of the inferred or measured
$PM_{2.5}$ mass concentration ($M_m$) to the actual $PM_{2.5}$ mass concentration ($M_a$). An $M_m/M_a$ ratio
of greater than one implies we are over-predicting the $PM_{2.5}$ loading, whereas a value less
than one implies we are under-predicting it.

## 3.1    Relative humidity and hygroscopic growth

One of the most widely discussed sources of error for OPS measurements is that caused
by water uptake (Crilley et al., 2018; Di Antonio et al., 2018; Malings et al., 2020; Wang et
al., 2015; Zheng et al., 2018). As relative humidity increases, hygroscopic particles (those
with non-zero hygroscopic growth parameters, $\kappa$) become larger as they take up water
(Petters and Kreidenweis, 2007), leading to an increase in scattering caused by their
increase in size. Additionally, water uptake changes the optical and chemical properties
of the aerosol (e.g., RI, density, etc.), which can complicate any corrections. The EPA
requires $PM_{2.5}$ measurements to be made at relative humidities between 30-40% (Chow
and Watson, 1998) to minimize the effects of hygroscopic growth on samples; however,
since very few low-cost OPS's do not control relative humidity (for example, with an in-
line dryer), this can often lead to errors when performing a calibration by co-location or
when comparing results between instrument types.



Figure 2 shows the impact the that RH can have on the accuracy of an OPS. There is little
effect until relative humidity reaches the deliquescence point of the aerosol, which
depends on aerosol composition. At higher relative humidities, OPS's will tend to
overestimate PM$_{2.5}$ mass, especially for aerosols comprised of hygroscopic materials.
When relative humidity approaches 95%, such overestimates in PM$_{2.5}$ mass become
exceedingly large: the OPC's observe a similar effect, with errors ranging from 100%-
500% depending on the hygroscopicity of the aerosol. Nephelometers see a more
pronounced effect with errors as high as 750% for extremely hygroscopic aerosols and
200%-300% errors for less hygroscopic aerosols.
The larger error of the nephelometer is caused in part by the fact that the PM$_{2.5}$ mass is
directly proportional to the total scattered light, which has no upper limit. For the OPC's,
particles that take up significant water can be assigned to larger size bins and thus will
not be integrated in the PM$_{2.5}$ mass calculation. At moderate humidities (50%-80%), errors
for both the nephelometers and OPC's can vary by as much as 20%-50%, which is in
agreement with a number of published experimental studies on the subject (Crilley et al.,
2018; Di Antonio et al., 2018; Malings et al., 2020; Zheng et al., 2018). In addition to
overestimating mass loadings at high relative humidity, the OPC's underestimate the
mass loadings when relative humidity is low. This is not caused by relative humidity, but
instead is a result of the "missing mass" below the detectable threshold of the OPC. The
low-cost OPC, which cannot detect particles smaller than 380 nm, misses between 30%-
90% of the mass, whereas the high-end OPC, which can detect particles larger than 100
nm, misses very little mass for most aerosol types. The only exception is the marine
aerosol, which has a refractive index that is substantially different than the aerosol with
which the instrument was calibrated.

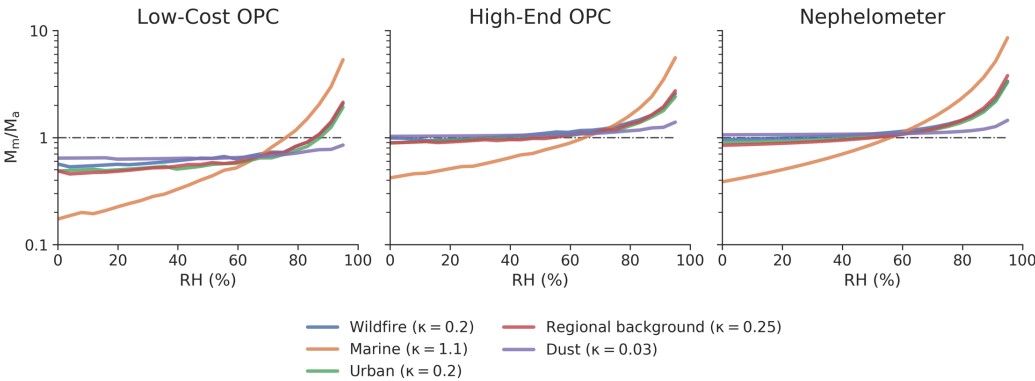

**Figure 2.** The accuracy in PM$_{2.5}$ mass loading for a given particle sensor (M$_m$/M$_a$) as a function of relative humidity, for common aerosol types. All three particle sensors were calibrated with ammonium sulfate (number-weighted Geometric Mean (GM) = 200 nm, Geometric Standard Deviation (GSD) = 1.65). Details on the physical and optical properties of the various aerosols can be found in Table 2.

## 3.2 Choice of calibration material and aerosol optical properties

OPC's are calibrated by correlating the scattering amplitude of known particle sizes for a particles of a given composition (Gao et al., 2013). The relationship between scattering amplitude and bin assignment (i.e., particle size) is heavily dependent on the aerosol's complex refractive index (RI). Figure 3 shows the Mie scattering curve for a range of common calibration materials, including both absorbing and non-absorbing materials. For a given particle size, the RI of the particle can result in a range of scattered light intensities (C$_{scat}$) that vary by as much as an order of magnitude. This can have pronounced effects on the calculated size (and hence mass) of a particle. In particular, the Mie curve for black carbon (BC) is substantially different than those of non-absorbing materials. As a result, for an OPC calibrated with a non-absorbing material (such as PSL's), smaller BC particles (diameters < 300 nm) will be overestimated in size, whereas larger BC particles (> 300 nm) will be underestimated. Even small changes in the scattering (real) component of the RI of the calibration material can lead to particles being assigned to the incorrect



bin: an RI higher than that of the calibration material will generally cause particles to be
assigned to bins that are too large (overestimating in size and mass), and an RI lower than
that of the calibration material will generally cause particles will be assigned to bins that
are too small (underestimating in size and mass). Considering that bins are often at least
hundreds of nm in width, the impact of such bin mis-assignment on reported mass can
be large. For both OPC's and nephelometers, this will lead to large errors in inferred
mass, though it can be more pronounced for OPC's, since the error for nephelometers is
proportional to the increase in scattering and is not affected by the mis-assignment of
individual particles to a particular size bin.

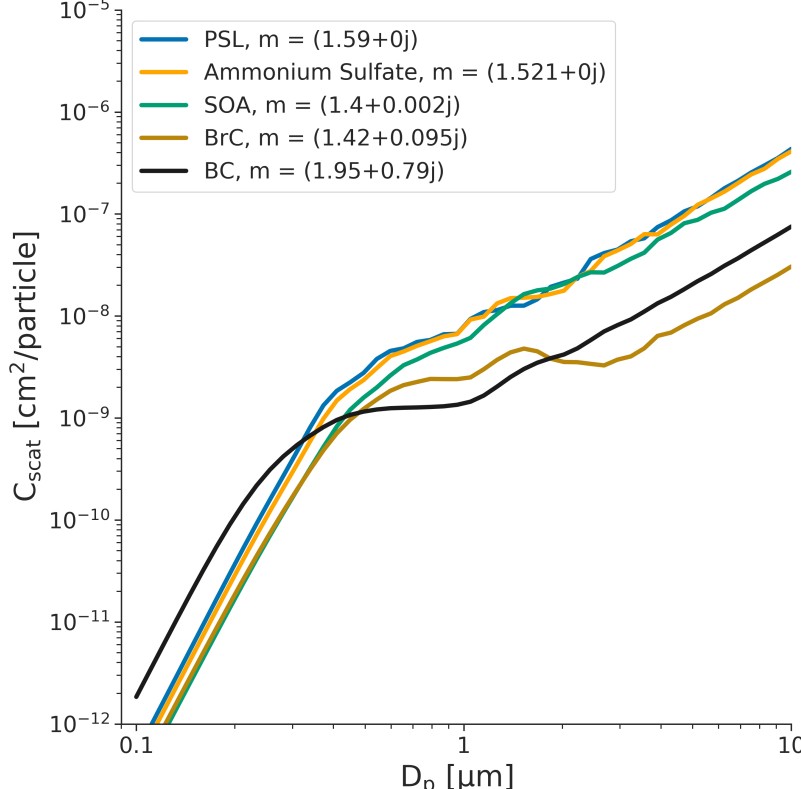

**Figure 3.** Mie curves (integrated over a viewing angle of 32°-88°) for a select group of common
calibration materials. Materials shown include polystyrene latex spheres (PSL's), ammonium
sulfate, secondary organic aerosol (SOA), and black carbon (BC). Small differences in the



refractive index of a measured material can lead to drastic bin mis-assignment, depending on
where bin boundaries are set at the time of calibration.
The effect of differences in refractive index on inferred PM$_{2.5}$ mass measurements is shown
in Fig. 4. Results are shown for a single aerosol distribution, in which the only parameter
allowed to vary is the RI. The real component of the refractive index is shown on the x
axis, with the upper and lower bounds being determined by the imaginary part of the
refractive index; the imaginary component ranges from 0 (non-absorbing) to 0.79 (black
carbon). The nephelometer (blue swatch in Fig. 4) is calibrated using ammonium sulfate
(m = 1.592 + 0j). When the nephelometer is evaluated at this exact RI (and a constant size
distribution), it measures mass accurately ($M_m/M_a$ = 1). However, if the real component of
the aerosol being evaluated is higher than that of the calibration standard, the total
scattering is greater, resulting in the inferred PM$_{2.5}$ mass being larger than the actual PM$_{2.5}$
mass ($M_m/M_a$ > 1). Similarly, as the absorbing component becomes larger, less of the
incoming light is scattered, resulting in a substantial underestimation of the mass loading.

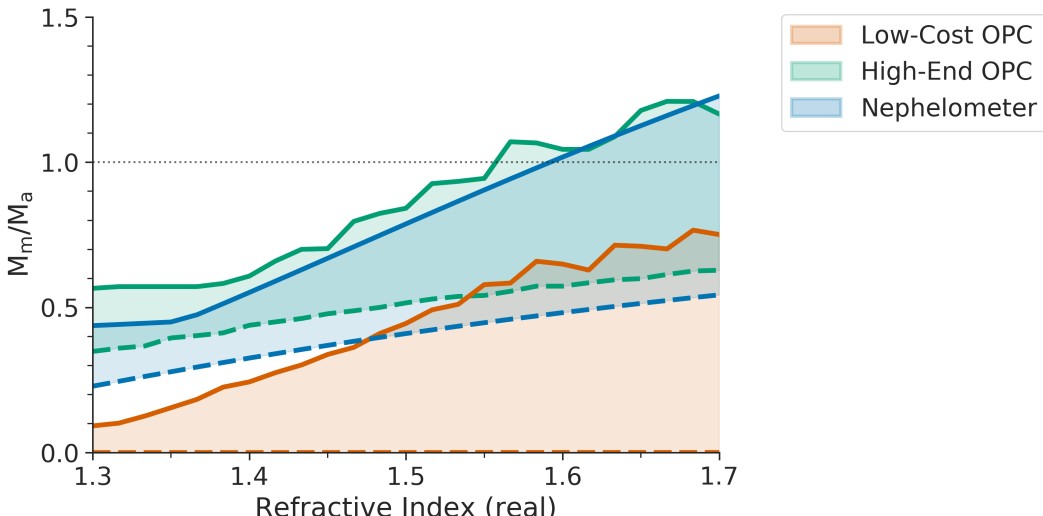

**Figure 4.** The accuracy of OPS's as a function of the refractive index of the aerosol being measured. The real component of the RI is on the x axis, and the width of each swatch bounded by the absorption/imaginary component, which spans from 0 (non-absorbing, solid line) to 0.79 (black carbon, dashed line). Results are shown for a nephelometer (blue), and the two OPC's (orange and green). All results are for a generic particle size distribution with number-weighted GM=200 nm and GSD=1.65 and the OPC's were calibrated with PSL's. A dotted line depicting the real part of the refractive index of the calibration material is also shown.

Also shown are the results for two OPC's. The high-end OPC (green) is sensitive to particles as small as 100 nm, whereas the low-cost OPC (red) is sensitive to particles as small as 380 nm. As the absorbing component of the refractive index becomes larger, the scattering amplitude across the entire distribution is too small for the OPC to detect, resulting in a mass reading of zero. Both OPC's exhibit this effect, but for the high-end OPC, fewer particles will fall below the size cutoff of the OPC than for the low-cost OPC, resulting in a less dramatic underestimation of the mass. Most commercially-available OPC's are more similar to the low-cost OPC, with lower limits of detection of around 500 nm. If operating in an environment where the aerosol is strongly absorbing, large underestimates in PM$_{2.5}$ should be expected. Even under conditions where the aerosol is not absorbing, the low-cost OPC largely underestimates the mass due to its high



minimum size cutoff. For nephelometers, the errors are not as drastic, but still do depend
strongly on the RI of the calibration aerosol used.
## 3.3      Changes in the Particle Size Distribution (PSD)
The ability of optical particle sensors to adapt to perturbations in the underlying particle
size distribution (PSD) is important because PSD's can be highly variable over short
periods of time, especially in urban areas with highly varying contributions from various
local sources. Fig. 5 shows the accuracy of all three OPS's as the function of the PSD of
the particles being measured. These calculations assume a single lognormal mode with
all other properties of the aerosols (density, refractive index, and hygroscopicity) held
constant. For the purpose of the model, the OPC's were calibrated using PSL's at each
bin boundary, and the nephelometer was calibrated using ammonium sulfate (N=1e4 cm$^{-3}$,
GM=400 nm, and GSD=1.65). The entire population of ammonium sulfate particles is
then evaluated while varying the number-weighted mean particle diameter (GM) and the
width of the distribution (GSD). For each PSD, we compute the relative accuracy of each
device and plot the results in Fig. 5, in which the color and contours correspond to the
$M_m/M_a$ metric.
The nephelometer substantially underestimates the mass concentration (by 50%-70%) for
most PSD's, since it is calibrated to a single PSD. As the PSD changes, the ratio of total
scattered light to integrated mass changes, causing the accuracy to change as well.
OPC's are potentially better since they measure the size of the particles and can
theoretically account for changes in the PSD; however, they are still subject to errors given
their limitations in detected size range. In particular, the low-cost OPC considerably
underestimates the mass (by 60%-90%) for most PSD's as the bulk of the mass is below
the detectable size limit of the OPC. As the geometric mean diameter increases in size,
or the width of the distribution becomes larger, a larger fraction of the particles enters
the detectable range, slightly improving the results for the low-cost OPC. The high-end
OPC is most able to adapt to the changes in the PSD due to its significantly smaller $d_{min}$
(100 nm); there is roughly a 20% difference across the entire range of PSD's shown. Unlike
the low-cost OPC, a majority of the mass falls within the detectable range of the high-
end OPC, resulting in little to no effect of changes to the PSD on accuracy of the mass
concentration measurement.

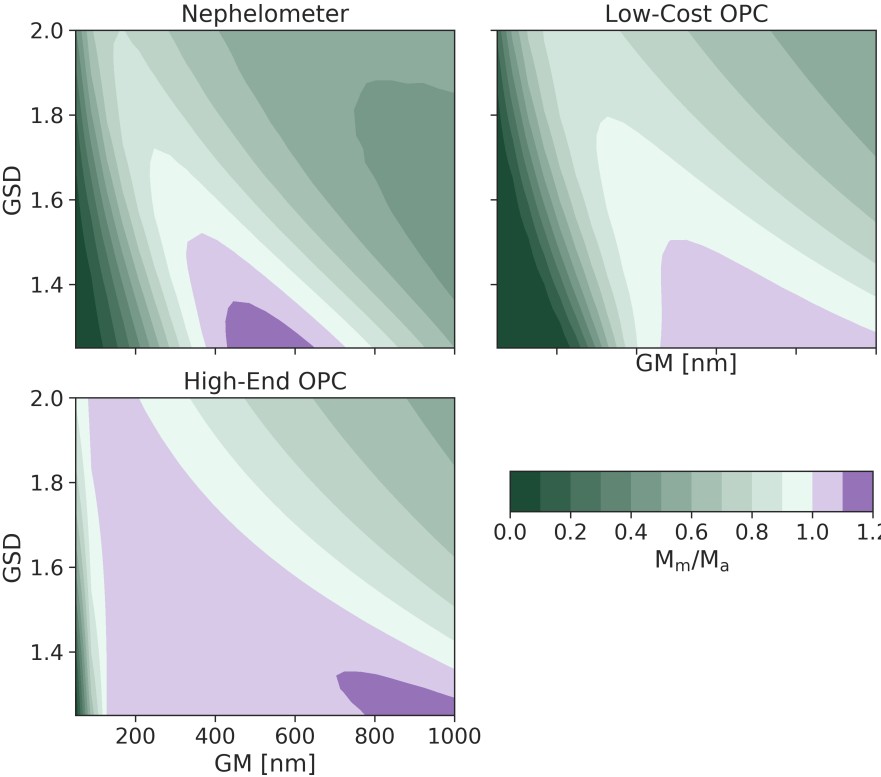

**Figure 5.** Mass concentration accuracy ($M_m/M_a$) of OPS's for a range of particle size distributions
(PSDs). Accuracy is shown for all combinations of PSD's with number-weighted geometric mean
diameters (GMs) between 100 - 1000 nm and geometric standard deviations (GSDs) between 1.2
– 2.0.  Perturbations in the PSD can lead to large errors for nephelometers and optical particle
counters with high minimum particle size cutoffs. All results are shown for ammonium sulfate
particles; the OPCs were calibrated with PSLs and the nephelometer was calibrated with
ammonium sulfate (N=1e4 cm$^{-3}$, GM=400 nm, and GSD=1.65).



While previous work has highlighted the importance of the varying PSD and its effect on making accurate mass measurements with OPS's (Di Antonio et al., 2018; Gao et al., 2013; Malings et al., 2020), the effect of 'missing mass' – the mass below the lowest size bin of an OPC – has received relatively little attention. The standard way to treat this missing mass is to empirically correct via regression analysis (Dacunto et al., 2015; Malings et al., 2020). While this can mitigate absolute errors, it requires the assumption that the PSD is constant in shape, varying only in magnitude. With particle loadings mostly below 10's of µg m$^{-3}$ throughout the United States, this assumption is unlikely to be a large source of absolute error. However, if the same approach were used in highly polluted environments where sub-300 nm aerosol loadings can easily reach hundreds of µg m$^{-3}$ (Bhandari et al., 2020; Gani et al., 2019), changes in the PSD are likely to lead to large errors (in both an absolute and relative sense) in mass loading measurements. Overall, nephelometers and OPC's with high minimum size cutoffs are prone to substantial uncertainties as the underlying PSD changes, whereas for OPC's with low minimum size cutoffs this effect is relatively minor.

## 4.    Implications and future work

In this work, we have laid out a framework for understanding the sensitivity of low-cost optical particle sensors to the various physical and optical properties of aerosols. We described a new Mie theory-based software package (*opcsim)* for modeling the response of OPS's to various aerosols and demonstrated its use for better understanding the strengths and limitations of various low-cost particle sensors. We also used the model to investigate how various potential pitfalls (e.g., changes to environmental conditions, mismatches between calibration particles and particles being measured) may contribute



to errors in mass concentration measurements. A summary of these results is given in
Table 3.
**Table 3**. Effects of changing environmental/aerosol parameters on the relative error in
measured mass loading by different OPS types.

| Parameter changed | OPS type | | |
|---|---|---|---|
| | **Low-Cost OPC** | **High-End OPC** | **Nephelometer** |
| RH & Hygroscopicity (Figure 2) | Very high for (20-200%) for hygroscopic materials when RH > ~75% | | |
| Optical Properties (Figure 4)[1] | Very High (30 – 100%) | Medium (20 – 60%) | Medium (20 – 75%) |
| Particle Size Distribution (Figure 5) | Very High 60 - 90% | Low < 20% | High 50 – 70% |

[1] Primarily a source of error when an OPS calibrated with non-absorbing particles
measures absorbing particles (or vice versa)
Consistent with previous studies, our results suggest that relative humidity is a large
source of uncertainty for all OPS's when the aerosol is hygroscopic and relative humidities
are above the deliquescence point, typically around 75%; additionally, the error
introduced by relative humidity is highly sensitive to the aerosols' affinity for water. This
is correctable, at least to first order, limiting the impact of RH error on final results (Crilley
et al., 2018; Di Antonio et al., 2018; Malings et al., 2020).  We showed that the aerosol
optical properties are most important for low-cost OPC's and of medium importance for
high-end OPC's and nephelometers. This is especially relevant when the aerosol is
strongly absorbing, as the amount of scattered light can make small particles
undetectable with inexpensive optical detectors. If it were possible to measure some
proxy for aerosol composition, it would be possible to vastly reduce this error and



improve the accuracy of mass measurements using OPS's. Finally, we showed the
underlying particle size distribution is very important for the accuracy of low-cost OPC's
and nephelometers, while being of low relative importance for high-end OPC's that can
properly count and size particles at low sizes. The ability of a given OPC to measure small
particles is found to be important, with marginal improvements leading to large gains in
ability to accurately infer mass. Additionally, the choice of calibrant is found to be
extremely important for both nephelometers and OPC's. Ensuring that OPS's are
calibrated intelligently (i.e., using particles similar to the aerosol to be detected) can lead
to significant improvements in expected performance. Additionally, the bin boundary
definitions for an OPC are also important, as defining them with large overlap in expected
$C_{scat}$ values can lead to significant bin misassignment and therefor incorrect mass
calculations.
Table 4 summarizes these results within the context of measurements of representative
real-world aerosol types. It provides an overview of the potential errors associated with
different types of optical particle sensors under various scenarios, with recommendations
for the type of calibration particles that would minimize errors in $PM_{2.5}$ mass
measurements. Generally, in environments where small particles (< 300 nm) comprise a
large percentage of the total mass, low-cost OPC's will be subject to considerable error.
This will also be the case in environments with substantial levels of light-absorbing
aerosol, such as wildfires or soot-heavy environments. (Sensor calibration using
absorbing particles could help mitigate this effect, though this would introduce new
errors when measuring non-absorbing aerosol.) In environments in which the underlying
aerosol size distribution is highly variable, such as urban environments or evolving
wildfire plumes, nephelometers and low-cost OPC's will struggle to keep up with the



changes in the relationship between the total scattered light and mass loading, leading
to large variance in the mass estimates.
The estimates and recommendations given in Table 4 are not intended to be
comprehensive, but rather serve as a starting point for characterizing the strengths and
limitations of low-cost OPS's using Mie theory (and specifically the *opcsim* software
package). Additional *opcsim* simulations carried out across a range of sensor designs,
calibrant particles, and measured particle types could provide more comprehensive and
quantitative estimates of errors in measured particle sizes and mass loadings, including
for individual sensors and individual use-cases. Future improvements to *opcsim* could be
made to allow for the simulation of more complex aerosols (e.g., externally-mixed
populations, other particle morphologies) or the inclusion of more complex bin-
assignment algorithms; comparison with laboratory studies (in which $M_m/M_a$ is measured
rather than just estimated) would also be useful. It is hoped that the Mie-theory-based
approach described here will lead to an improved understanding of the errors associated
with low-cost optical PM measurements, insight into calibration techniques that minimize
such errors, and ultimately guidance into the design of new PM sensors for improved low-
cost measurements of air quality and human exposure.



Table 4. Summary of expected performance and recommendations for calibration materials for use of low-cost optical particle sensors to measure different aerosol types.

| AEROSOL TYPE | AEROSOL PROPERTIES | SUGGESTED CALIBRANT | SENSOR PERFORMANCE BY OPS TYPE [1] | | |
|---|---|---|---|---|---|
| | | | Low-Cost OPC | High-End OPC | Nephelometer |
| FOSSIL-FUEL COMBUSTION | Very small PSD, mostly non-hygroscopic, moderate absorbing RI | Calibrate with aerosols closer in RI, such as from combustion sources | Will perform poorly due to the small PSD and absorption component of the aerosol | Will perform moderately well though will miss ultrafine particles | Can perform moderately well if calibrated using appropriate materials |
| WILDFIRE | Varying PSD, moderate absorbing component of RI | Calibrate with aerosol of similar optical properties and PSD (ideally biomass smoke) | Will likely undersize and underestimate mass due to the absorbing component of the aerosol; the PSD will change with proximity to the source leading to changes in accuracy | Will perform moderately well, though may mis-size the particles as the properties of the aerosol change as the plume evolves | Can perform well under certain circumstances; moderate error should be expected as the PSD of the wildfire plume evolves |
| URBAN | Varying PSD, moderate hygroscopicity | Calibrate with NIST urban aerosol | Performance depends on uniformity of sources; large errors will occur as aerosol source (and PSD) changes | Will perform moderately well to well, though will miss ultrafine particles | Will perform moderately well if averaged over a long period of time to normalize the PSD |
| DUST | Large PSD, non-hygroscopic | Calibrate with Arizona Road dust | Likely to perform well, given the large particle sizes | Likely to perform well, given the large particle sizes | Likely to perform well |

[1] Based on properties of the aerosol only, and not external environmental parameters (i.e., RH).





Acknowledgments
This work was funded by the Tata Center for Technology and Design at the Massachusetts
Institute of Technology, as well as the US Environmental Protection
Agency under assistance agreement RD-83618301. It has not been formally reviewed
by the EPA; EPA does not endorse any products or commercial services mentioned in
this publication. Additionally, the authors would like to thank Eben Cross and Timothy
Onasch for helpful conversations regarding the operating principles of these sensors,
Colette Heald for helpful suggestions regarding the summary of the paper, as well as the
Kroll and Heald research groups at MIT for thoroughly testing the *opcsim* software
package.



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
