# Peer review of "Assessing the accuracy of low-cost optical particle"

_Atmospheric Measurement Techniques, 2020_

## Referee Comment (RC1) · Anonymous Referee #1 · 17 Jul 2020

This work by Hagan and Kroll presents an open source model, opcsim, based on mie theory that they suggest can be used to evaluate the ability of low-cost optical particle sensors (optical particle counters and nephelometers) to accurately characterize the size distribution and/or mass loading of aerosol particles. The authors use this model to evaluate the ability of different sensor technology to measure PM2.5 mass concentration and the effect of RH, aerosol composition and size distribution on these sensors. My concerns relate more to the authors use of the model to evaluate low cost-sensors and their conclusions from it. I realise that the authors did not want to be too specific (e.g. focusing on one particular sensor), but I did find that the findings are quite broad. I think this is best highlighted by their conclusion that the lower particle

size cut off is critical, especially for low-cost particle sensors, as this can be relatively high (ca. 500nm). To me this is kind of obvious, even the very best OPC will not be able measure particles below their lower size bin. Consequently, it not surprising that this is a large source of error for a low cost OPC relative to an OPC that has a lower size cut off. In my opinion, the results from the model simulation seemed to be overly affected by the lower size cut off chosen for the low and high cost OPC. I further detail some of more these concerns in the specific comments below.

Overall, in my opinion, the paper is well written, clearly presented and the model will be of interest and use to the community to evaluate new sensors and their potential errors prior to lab testing.

Specific comments

Page 20, line 19: The authors state 'Even under conditions where the aerosol is not absorbing, the low-cost OPC largely underestimates the mass due to its high minimum size cutoff' If the errors are associated more with the size cut off, then how can you make statements about the effect of aerosol refractive index on the low-cost sensor?

Page 22, line 5: Is this issue with this comparison that the chosen model aerosol mostly falls below the size detection limit of the low cost OPC (is 500nm)? For me, when I look at Fig 5, when the GM of the aerosol is above 500nm the low cost OPC performs well. I do not quite see the point of this simulation, as these uncertainties related to lower size cut off are inherent for any OPC, irrespective if they are low or high cost?

Page 23. Line 8-10: Could you provide an estimation of the absolute error for low particle mass concentrations from the literature?

Page 25, line 25: For urban environments where the PSD is changing, could this error in low-cost OPC be mitigated by sampling for a longer time period? For to put it another way, is this error dependent on sampling time resolution of the OPC?

Table 4: These NIST standards are very expensive, perhaps you could suggest

cheaper alternatives?

---

## Referee Comment (RC2) · R. Subramanian (Referee) · 22 Jul 2020

R. Subramanian (Referee)

subu@lisa.u-pec.fr

Low-cost sensors (LCSs) are widely used now-a-days for objectives ranging from citizen awareness to scientific research. Many studies have been conducted to evaluate commercially-available LCSs (including by my group), but systematic design-based analyses have been lacking. The recent He et al. (2019) https://doi.org/10.1080/02786826.2019.1696015 and this submission are important steps towards overcoming that shortcoming.

Some specific comments are listed below, but a broad comment is that the results as presented could be made more useful with simulations that consider widely-used

commercial devices.

- For example, common high-end OPSs (TSI 640, Grimm 180) do not measure down to 100 nm, but rather to 180-200 nm. (The Handix POPS - a really nice instrument! I want one! - is not commonly used for ambient PM measurements and probably gets overwhelmed by ambient PN outside of balloon flights.) I suspect this higher cutoff will significantly worsen the performance of the high-end OPC presented here, but it will be more realistic.

- The nephelometer collection angle used here is 7-173 degrees, which may be true for expensive TSI-type nephelometers, but such data are not available for LCSs (as shown in Table 1). The best data might be from Kelly et al. (2017) http://dx.doi.org/10.1016/j.envpol.2016.12.039, who say the Plantower collects scattered light at 90 degrees and the Shinyei at 45 degrees. It might help if the nephelometer results presented in this manuscript instead used 90+/-45 degrees or a similar range (or maybe the Plantower company can provide that information?)

- Finally, a key difference between the results presented here and sensor performance in the real world is that manufacturers may calibrate the sensor to ambient urban aerosol. This appears to be the case for Plantower - we were told (Malings et al. 2019) they calibrate to reference monitors in Chinese cities. (Met-One NPM is calibrated with 600 nm PSL.) Would it be possible to use a "typical" Beijing PSD instead of the ammonium sulfate calibration basis in the nephelometer results presented here?

I admit some of this may be more complicated and a lot more work, but people wanting to use the opcsim software may invariably want to do just that... And might turn to Dr Hagan for assistance anyway! Might as well get ahead of the curve and also get it published.

Specific comments: Page 15, line 8: what is "actual PM2.5 mass"? Is that the mass at 35% RH (like EPA regulations)? Please specify.

Page 16, lines 18-20: wouldn't the "missing mass" problem be present at all humidities, just (over)compensated at higher RH due to hygroscopic growth?

Figure 2: Some of these results are hard to decipher on a log scale. Maybe show it on a linear scale? (The marine case seems to be an extreme/can be moved to SI?)

Also in Fig 2, at least in Malings et al. (2019), we showed errors in the as-reported Plantower data as a function of RH (Table S2). Perhaps a more RH-resolved comparison could be made?

Page 18, lines 4-6: Can RI changes really cause such significant mis-assignment? Maybe at the margins/bin boundaries, sure. But the PSL, SOA, ammonium sulfate Mie curves (Fig 3) are pretty close to each other, and these are the dominant PM2.5 components by mass and (probably, OPC-wise anyway) number.

Page 19, line 9: "calibrated using PSL", not ammonium sulfate? (PSL matches the RI in parentheses and the caption of Fig 4.)

Pages 24-25: "some proxy for aerosol composition" - In Malings et al. (2019), we used PM composition data from the EPA CSN network (139 sites across the US, Fig S4) with a wide range of chemical composition and found the results did not change significantly (Fig S5), so long as some fRH correction was used. Might be relevant here.

Page 25, line 11: "therefore" not "therefor"

Page 25, line 24: are urban OECD aerosol size distributions "highly variable"? This might be another example of "errors that may not be significant in the US/EU but are important in developing countries".

Table 4: some issues with this:

- "small PSD" or "large PSD" suggests narrow or broad size distribution, but I suspect the authors mean "smaller aerosols" or "larger aerosols". Please clarify.

- NIST urban aerosol - SRM 1648 was collected in 1976-1977 and may not be representative of, well, anything these days...

- Maybe the "aerosol properties" column should have some references?

- The authors emphasize this table is only a start, so it's fine to include it, I guess. There could be an entire workshop devoted to Table 4...

Throughout: Malings et al. (2019), not Malings et al. (2020). (I don't know why T&F's "download citation" shows the year as 2020. The paper was published in June 2019 and my downloaded PDF shows 2019 as the year for citation.)

Throughout: "The pluralization of abbreviations, too, requires no apostrophes. More than one CD = CDs... Etc." - Benjamin Dreyer, Dreyer's English (2019). See also: https://www.chicagomanualofstyle.org/qanda/data/faq/topics/Plurals.html?page=1

---

## Referee Comment (RC3) · Anonymous Referee #3 · 3 Aug 2020

This article uses Mie scattering theory to evaluate accuracy of three classes of optical particle sensors used in aerosol measurements: nephelometers, low-cost particle counters with limited size range detectability, and high end particle counters with wider size range, especially at the lower end of size spectrum. Manuscript is well written and addresses important factors affecting accuracy of particle measurements with an emphasis on mass loading calculations. Authors show the importance of three factors in evaluating accuracy of results from different optical sensors: particle growth in presence of humidity; difference between optical properties of the measured particles and those used for calibrating the instrument; and difference between the size range of measured particles with the calibration particles. My major concern about this article

is that it is rather incomplete at the present state. Following are a couple of ways in which I think this study can be improved.

1. More analysis needs to be done to demonstrate the importance of these factors in measuring real world particles. Examples presented in the text are simple hypothetical cases that just show the importance of each factor. It is important for the reader to see how much the results of actual atmospheric particle measurements get affected by these errors. Analysis of data from previous studies would be helpful to obtain a picture of how much those data would change if corrected for the effects. One major question is to see if the final conclusion from some of previous studies would be affected based on these factors.

2. The abstract and introduction create the expectation that this article would also address the interaction between different factors and the impact on accuracy of measurement results. However, presented cases are all single-factor studies. If authors believe the interdependence of these factors is significant, these mutual effects should be addressed in the paper.

Technical Corrections:

Title – "low-cost" word can be omitted from the title as this manuscript looks at low cast as well as "high end" instruments.

P5, L3 – Please provide a brief description of factors resulting in "changes in aerosol optical properties". P8, Table 1 – Have the authors tried contacting the manufacturers to obtain the data that is missing in the literature?

P12, L7 – Why is geometric mean used for mass calculations? Are the authors suggesting that they use the measured pulse heights from individual particles rather than assuming uniform size distribution across each individual size bin? If yes, what is the point of referring to size bins?

P12, L24 – Similar comment as above. Please clarify the reason for preferring geomet-

ric mean diameter.

P15, L9-10 –"overestimate" and "underestimate" would be more accurate terms for these sentences.

P15, L22-24 – The wording in this section is unclear. Do the authors mean that "very few low cost OPSs control relative humidity…"?

P16, L1- Should read "Figure 2 shows the impact that RH can…"

P17, L20 – Please add more explanation about Mie scattering of black carbon particles in the <300nm range. The trend shown in >300nm range is understandable based on the BC particles being more absorbing, but an explanation for <300nm range is missing.

Fig 5 – Please show the contour line corresponding to Mm/Ma=1

Fig 5 – It seems that even at the calibration conditions (GM=400 nm, and GSD=1.65), mass loading accuracy is not equal to unity in any of the plots. What is the explanation?

Table 4 – Need to mention that the main metric in consideration in this table is mass loading. Clearly, if the focus in a study is on evolution of particle size distribution rather than integral mass loading, nephelometer can't be recommended.

---

## Author Comment (AC1) · 17 Sep 2020

**Response to Reviewers**

We thank the reviewers for their comments. We have worked to address all concerns as discussed below, with a number of changes made to the manuscript and SI. We believe such changes increase the robustness of the conclusions, and generally have led to an improved manuscript.

Responses to specific comments are below. The original reviewer comment is in italics with the author response in normal font.

**Reviewer 1**

**Major Comments**

This work by Hagan and Kroll presents an open source model, opcsim, based on mie theory that they suggest can be used to evaluate the ability of low-cost optical particle sensors (optical particle counters and nephelometers) to accurately characterize the size distribution and/or mass loading of aerosol particles. The authors use this model to evaluate the ability of different sensor technology to measure PM2.5 mass concentration and the effect of RH, aerosol composition and size distribution on these sensors. My concerns relate more to the authors use of the model to evaluate low cost-sensors and their conclusions from it. I realise that the authors did not want to be too specific (e.g. focusing on one particular sensor), but I did find that the findings are quite broad. I think this is best highlighted by their conclusion that the lower particle size cut off is critical, especially for low-cost particle sensors, as this can be relatively high (ca. 500nm). To me this is kind of obvious, even the very best OPC will not be able measure particles below their lower size bin. Consequently, it not surprising that this is a large source of error for a low cost OPC relative to an OPC that has a lower size cut off. In my opinion, the results from the model simulation seemed to be overly affected by the lower size cut off chosen for the low and high cost OPC

As the reviewer mentions, we were intentionally broad, not showing results for a specific sensor model. This is by design, as there are simply too many specific sensors to complete an analysis for each individual one. This is why this model is open-sourced, with extensive documentation, is so that anyone can run the analysis for a specific sensor model. There is documentation with the model code (https://github.com/dhhagan/opcsim) showing how to do this for the Alphasense OPC-N2 – a generally available low-cost optical particle counter. In addition, we agree that "the very best OPC will not be able to measure particles below their lower size bin". However, as a survey of the low-cost sensor literature will show, (1) this is not generally appreciated amongst this community; and (2) it has not been shown just how much of an impact this can have when making PM measurements using these devices across a wide range of environments and environmental conditions.

In order to clarify the objectives of this work, and specifically to explain why individual sensors are not systematically investigated, we have added the following text to the introduction:

The following text has been appended to the introduction:

The objective of this work is to describe the model and software and to investigate broad influences on aerosol properties and sensor parameters on measurement performance. This present work does not investigate the performance of individual commercially available sensors under the full range of conditions expected in the atmosphere; but such studies are enabled by this modeling tool and are an important future extension of this work.

Overall, in my opinion, the paper is well written, clearly presented and the model will be of interest and use to the community to evaluate new sensors and their potential errors prior to lab testing.

We thank the reviewer for their assessment and address the individual comments below.

**Minor Comments**

Page 20, line 19: The authors state 'Even under conditions where the aerosol is not absorbing, the low-cost OPC largely underestimates the mass due to its high minimum size cutoff' If the errors are associated more with the size cut off, then how can you make statements about the effect of aerosol refractive index on the low-cost sensor

We are able to make these comments because we perform two separate experiments: (1) we hold the aerosol optical properties constant and change only the size distribution (Section 3.3); (2) we hold the size distribution constant and change only the aerosol optical properties (Section 3.2). This allows us to separate out the impact of each factor individually.

Page 22, line 5: Is this issue with this comparison that the chosen model aerosol mostly falls below the size detection limit of the low cost OPC (is 500nm)? For me, when I look at Fig 5, when the GM of the aerosol is above 500nm the low cost OPC performs well. I do not quite see the point of this simulation, as these uncertainties related to lower size cut off are inherent for any OPC, irrespective if they are low or high cost?

The reviewer is correct that these issues are inherent for any OPC irrespective of cost. However, the purpose of this figure is to demonstrate the relative ability of different OPSs to accurately measure varying size distributions. Yes, an OPC will do a good job for an aerosol distribution with a number-weighted GM ~ 500nm; however, there are very few instances where you might encounter a distribution that large in the real-world. Figure 5 shows that as the size cutoff of the OPC improves, the ability of the OPC to more accurately size and count particles improves for a wider range of size distributions. Importantly, it also demonstrates that a nephelometer cannot adapt to changes in size distribution well at all, which is important because many low-cost, widely used AQ networks are based on the Plantower PMS5003 sensor, which is a nephelometer.

Page 23. Line 8-10: Could you provide an estimation of the absolute error for low particle mass concentrations from the literature?

It is difficult to report estimates for the absolute error as it depends largely on the exact area that data was collected and can be heavily influenced by the normalization of the size distribution via longer averaging times. The estimate for the error can be computed if the size distribution is known. A statement has been added to the conclusion that summarizes the need for improved co-location data with size-resolved measurements so that these numbers can be better understood: "Additionally, co-located data with size-resolved measurements would allow for improved validation of the OPC component of this model." (P 27, lines 6-8)

Table 4: These NIST standards are very expensive, perhaps you could suggest cheaper alternatives?

We have changed this column to include collected samples; these would be less standardized and less well-characterized, but also are substantially less expensive than the NIST standards listed.

**Reviewer 2**

**Major Comments**

Low-cost sensors (LCSs) are widely used now-a-days for objectives ranging from citizen awareness to scientific research. Many studies have been conducted to evaluate commercially-available LCSs (including by my group), but systematic design-based analyses have been lacking. The recent He et al. (2019) https://doi.org/10.1080/02786826.2019.1696015 and this submission are important steps towards overcoming that shortcoming. Some specific comments are listed below, but a broad comment is that the results as presented could be made more useful with simulations that consider widely-used commercial devices.

Thank you for these comments. We completely agree that more device-specific comparisons would be very useful! First, however, we felt it was important to first (1) describe the approach used, (2) make the modeling tools available, and (3) investigate general considerations (related to aerosol properties and sensor parameters) related to measurement accuracy. In particular, we aimed to show that the size range for which the device can measure is very important as opposed to comparing specific instruments. We do have follow-up work planned that both compares the results of this model to field and

laboratory measurements by specific nephelometers and OPCs; but our hope is that this tool may also be useful to others to investigate sensor performance for specific sensors and/or aerosol types.

For example, common high-end OPSs (TSI 640, Grimm 180) do not measure down to 100 nm, but rather to 180-200 nm. (The Handix POPS - a really nice instrument! I want one! - is not commonly used for ambient PM measurements and probably gets overwhelmed by ambient PN outside of balloon flights.) I suspect this higher cutoff will significantly worsen the performance of the high-end OPC presented here, but it will be more realistic.

This is a very good point and comes down to the definition of 'high-end' – whether it's the best one possible or the best one widely available/used. We chose the former, as a rough estimate of the smallest particle size that an optical sensor could reasonably measure. To make this clear, we now discuss this in the text: "We note that many expensive OPCs cannot measure particles down to 100 nm; this lower size cutoff was chosen as an approximate smallest particle size that an optical sensor can detect." (P 14, lines 11-12)

The nephelometer collection angle used here is 7-173 degrees, which may be true for expensive TSI-type nephelometers, but such data are not available for LCSs (as shown in Table 1). The best data might be from Kelly et al. (2017) http://dx.doi.org/10.1016/j.envpol.2016.12.039, who say the Plantower collects scattered light at 90 degrees and the Shinyei at 45 degrees. It might help if the nephelometer results presented in this manuscript instead used 90+/-45 degrees or a similar range (or maybe the Plantower company can provide that information?)

As the reviewer points out, we did not systematically investigate the effect of viewing angle, which can vary from sensor to sensor. We agree that the center of the collection angle is at 90° for the Plantower, though from a physical inspection it does appear there is no real focusing of the collected light. The Shinyei would likely be classified as a photometer, which is considered a subset of the nephelometer in this work.

We have computed results for two alternate optical setups, one with a collection angle of 90 +/- 45 degrees, and another with a collection angle of 45 +/- 5 degrees. These results are shown below. While the range for which a 1:1 value is obtained do change with collection angle (as expected), the results are qualitatively the same. A section on this has been added to the SI, along with a new Figure S4 (shown below); this is referred to in the main text as well (P22 lines 3-5).

Finally, a key difference between the results presented here and sensor performance in the real world is that manufacturers may calibrate the sensor to ambient urban aerosol. This appears to be the case for Plantower - we were told (Malings et al. 2019) they calibrate to reference monitors in Chinese cities. (Met-One NPM is calibrated with 600 nm PSL.) Would it be possible to use a "typical" Beijing PSD instead of the ammonium sulfate calibration basis in the nephelometer results presented here?

We chose polydisperse ammonium sulfate as a standard for comparison across all nephelometer types. A Plantower-specific study should definitely include Beijing urban aerosol as the calibrant. We are currently working on applying opcsim to the Plantower, but as noted above, focusing on specific sensors is beyond the scope of this work. I admit some of this may be more complicated and a lot more work, but people wanting to use the opcsim software may invariably want to do just that... And might turn to Dr Hagan for assistance anyway! Might as well get ahead of the curve and also get it published.

We agree and are working on specific use-cases; but we also didn't want to wait until that work was completed to describe (and release) the model. The number of specific sensors, aerosol types, and environmental conditions is too large to cover in a single study; so instead, we hope this can be useful to others to examine specific use cases. Each reader/user may want to simulate their sensor in a slightly different way with different calibrants and/or sensor specifications. To make this easier, extensive documentation showing exactly how to do this using this model can be found in the code's documentation (https://dhhagan.github.io/opcsim/). Questions pertaining to the code itself or to specific examples can be posted publicly on the GitHub repository itself (https://github.com/dhhagan/opcsim).

**Minor Comments**

Page 15, line 8: what is "actual PM2.5 mass"? Is that the mass at 35% RH (like EPA regulations)? Please specify.

This paper calculates 'actual' mass at 0% RH. The difference between mass at 0% and 35% RH is negligible for most aerosols. Per Figure 2, with the exception of marine aerosol, kappa values are 0.25 or less. At a kappa of 0.25, the expected difference in mass at 35% relative to 0% RH is ~10%. This has been clarified in the text. (P 15, lines 8-9)

Page 16, lines 18-20: wouldn't the "missing mass" problem be present at all humidities, just (over)compensated at higher RH due to hygroscopic growth?

Yes, the reviewer is correct. The missing mass problem is present at all humidities and is simply over-compensated at higher RH. The text has been updated (P 16 lines 17-19) to make this clearer: "In addition to overestimating mass loadings at high relative humidity due to hygroscopic growth, the OPCs underestimate the mass loadings across all relative humidities. This is not caused by relative humidity or a lack of hygroscopic growth, but instead is a result of the "missing mass" below the detectable threshold of the OPC."

Figure 2: Some of these results are hard to decipher on a log scale. Maybe show it on a linear scale? (The marine case seems to be an extreme/can be moved to SI?)

We considered this, for the exact reasons the reviewer states. However, the linear scale actually makes the data harder to interpret, as the values for different aerosol types appear closer together. The linear-scale image (without marine) is shown below for reference.

Thus, we prefer to keep the figure with a log axis, to highlight differences between different particle types.

Also in Fig 2, at least in Malings et al. (2019), we showed errors in the as-reported Plantower data as a function of RH (Table S2). Perhaps a more RH-resolved comparison could be made?

This was the primary objective of Figure 2 (which is expressed as relative error rather than absolute error). However, these are calculations on model aerosol and not measurements of real-world particles (as in Malings 2020), so we prefer not to tabulate results, which might give the wrong impression about the applicability of these errors to real-word systems/sensors.

Page 18, lines 4-6: Can RI changes really cause such significant mis-assignment? Maybe at the margins/bin boundaries, sure. But the PSL, SOA, ammonium sulfate Mie curves (Fig 3) are pretty close to each other, and these are the dominant PM2.5 components by mass and (probably, OPC-wise anyway) number.

This is an important point, which we did not address explicitly in the original submission. We have added an additional section and multiple figures to the SI that shows just how much of an impact slight changes in RI can have on the bin assignment for an OPC. This new section in the SI shows that yes, even small changes in RI can lead to large changes in how a particle is assigned to a bin – it can span many bins for a given particle size and depends strongly on the properties of the OPC as well as the aerosol itself. These text and figures are given below:

**Impact of changes in aerosol optical properties on OPC bin assignment**

Here we investigate how bin assignment may be affected If the aerosol optical properties of the measured aerosol are different than the material used to calibrate an OPC. We compute the light scattering for a range of possible refractive index values and plot them in Figure S2 below for a 600 nm particle. Across a wide range of optical properties, the particle can be placed in 3 different bins or left out completely ("NO BIN"). However, it should be noted that fairly large

---

## Author Comment (AC2) · 17 Sep 2020

**Response to Reviewers**

We thank the reviewers for their comments. We have worked to address all concerns as discussed below, with a number of changes made to the manuscript and SI. We believe such changes increase the robustness of the conclusions, and generally have led to an improved manuscript.

Responses to specific comments are below. The original reviewer comment is in italics with the author response in normal font.

**Reviewer 1**

**Major Comments**

*This work by Hagan and Kroll presents an open source model, opcsim, based on mie theory that they suggest can be used to evaluate the ability of low-cost optical particle sensors (optical particle counters and nephelometers) to accurately characterize the size distribution and/or mass loading of aerosol particles. The authors use this model to evaluate the ability of different sensor technology to measure PM2.5 mass concentration and the effect of RH, aerosol composition and size distribution on these sensors. My concerns relate more to the authors use of the model to evaluate low cost-sensors and their conclusions from it. I realise that the authors did not want to be too specific (e.g. focusing on one particular sensor), but I did find that the findings are quite broad. I think this is best highlighted by their conclusion that the lower particle size cut off is critical, especially for low-cost particle sensors, as this can be relatively high (ca. 500nm). To me this is kind of obvious, even the very best OPC will not be able measure particles below their lower size bin. Consequently, it not surprising that this is a large source of error for a low cost OPC relative to an OPC that has a lower size cut off. In my opinion, the results from the model simulation seemed to be overly affected by the lower size cut off chosen for the low and high cost OPC*

> As the reviewer mentions, we were intentionally broad, not showing results for a specific sensor model. This is by design, as there are simply too many specific sensors to complete an analysis for each individual one. This is why this model is open-sourced, with extensive documentation, is so that anyone can run the analysis for a specific sensor model. There is documentation with the model code (https://github.com/dhhagan/opcsim) showing how to do this for the Alphasense OPC-N2 – a generally available low-cost optical particle counter. In addition, we agree that "the very best OPC will not be able to measure particles below their lower size bin". However, as a survey of the low-cost sensor literature will show, (1) this is not generally appreciated amongst this community; and (2) it has not been shown just how much of an impact this can have when making PM measurements using these devices across a wide range of environments and environmental conditions.

In order to clarify the objectives of this work, and specifically to explain why individual sensors are not systematically investigated, we have added the following text to the introduction:

> The following text has been appended to the introduction:
>
> The objective of this work is to describe the model and software and to investigate broad influences on aerosol properties and sensor parameters on measurement performance. This present work does not investigate the performance of individual commercially available sensors under the full range of conditions expected in the atmosphere; but such studies are enabled by this modeling tool and are an important future extension of this work.

*Overall, in my opinion, the paper is well written, clearly presented and the model will be of interest and use to the community to evaluate new sensors and their potential errors prior to lab testing.*

> We thank the reviewer for their assessment and address the individual comments below.

**Minor Comments**

*Page 20, line 19: The authors state 'Even under conditions where the aerosol is not absorbing, the low-cost OPC largely underestimates the mass due to its high minimum size cutoff' If the errors are associated more with the size cut off, then how can you make statements about the effect of aerosol refractive index on the low-cost sensor*

> We are able to make these comments because we perform two separate experiments: (1) we hold the aerosol optical properties constant and change only the size distribution (Section 3.3); (2) we hold the size distribution constant and change only the aerosol optical properties (Section 3.2). This allows us to separate out the impact of each factor individually.

*Page 22, line 5: Is this issue with this comparison that the chosen model aerosol mostly falls below the size detection limit of the low cost OPC (is 500nm)? For me, when I look at Fig 5, when the GM of the aerosol is above 500nm the low cost OPC performs well. I do not quite see the point of this simulation, as these uncertainties related to lower size cut off are inherent for any OPC, irrespective if they are low or high cost?*

> The reviewer is correct that these issues are inherent for any OPC irrespective of cost. However, the purpose of this figure is to demonstrate the relative ability of different OPSs to accurately measure varying size distributions. Yes, an OPC will do a good job for an aerosol distribution with a number-weighted GM ~ 500nm; however, there are very few instances where you might encounter a distribution that large in the real-world. Figure 5 shows that as the size cutoff of the OPC improves, the ability of the OPC to more

accurately size and count particles improves for a wider range of size distributions. Importantly, it also demonstrates that a nephelometer cannot adapt to changes in size distribution well at all, which is important because many low-cost, widely used AQ networks are based on the Plantower PMS5003 sensor, which is a nephelometer.

*Page 23. Line 8-10: Could you provide an estimation of the absolute error for low particle mass concentrations from the literature?*

It is difficult to report estimates for the absolute error as it depends largely on the exact area that data was collected and can be heavily influenced by the normalization of the size distribution via longer averaging times. The estimate for the error can be computed if the size distribution is known. A statement has been added to the conclusion that summarizes the need for improved co-location data with size-resolved measurements so that these numbers can be better understood: "Additionally, co-located data with size-resolved measurements would allow for improved validation of the OPC component of this model." (P 27, lines 6-8)

*Table 4: These NIST standards are very expensive, perhaps you could suggest cheaper alternatives?*

We have changed this column to include collected samples; these would be less standardized and less well-characterized, but also are substantially less expensive than the NIST standards listed.

**Reviewer 2**

**Major Comments**

*Low-cost sensors (LCSs) are widely used now-a-days for objectives ranging from citizen awareness to scientific research. Many studies have been conducted to evaluate commercially-available LCSs (including by my group), but systematic design-based analyses have been lacking. The recent He et al. (2019) https://doi.org/10.1080/02786826.2019.1696015 and this submission are important steps towards overcoming that shortcoming. Some specific comments are listed below, but a broad comment is that the results as presented could be made more useful with simulations that consider widely-used commercial devices.*

Thank you for these comments. We completely agree that more device-specific comparisons would be very useful! First, however, we felt it was important to first (1) describe the approach used, (2) make the modeling tools available, and (3) investigate general considerations (related to aerosol properties and sensor parameters) related to measurement accuracy. In particular, we aimed to show that the size range for which the device can measure is very important as opposed to comparing specific instruments. We do have follow-up work planned that both compares the results of this model to field and

laboratory measurements by specific nephelometers and OPCs; but our hope is that this tool may also be useful to others to investigate sensor performance for specific sensors and/or aerosol types.

*For example, common high-end OPSs (TSI 640, Grimm 180) do not measure down to 100 nm, but rather to 180-200 nm. (The Handix POPS - a really nice instrument! I want one! - is not commonly used for ambient PM measurements and probably gets overwhelmed by ambient PN outside of balloon flights.) I suspect this higher cutoff will significantly worsen the performance of the high-end OPC presented here, but it will be more realistic.*

> This is a very good point and comes down to the definition of 'high-end' – whether it's the best one possible or the best one widely available/used. We chose the former, as a rough estimate of the smallest particle size that an optical sensor could reasonably measure. To make this clear, we now discuss this in the text: "We note that many expensive OPCs cannot measure particles down to 100 nm; this lower size cutoff was chosen as an approximate smallest particle size that an optical sensor can detect." (P 14, lines 11-12)

*The nephelometer collection angle used here is 7-173 degrees, which may be true for expensive TSI-type nephelometers, but such data are not available for LCSs (as shown in Table 1). The best data might be from Kelly et al. (2017) http://dx.doi.org/10.1016/j.envpol.2016.12.039, who say the Plantower collects scattered light at 90 degrees and the Shinyei at 45 degrees. It might help if the nephelometer results presented in this manuscript instead used 90+/-45 degrees or a similar range (or maybe the Plantower company can provide that information?)*

> As the reviewer points out, we did not systematically investigate the effect of viewing angle, which can vary from sensor to sensor. We agree that the center of the collection angle is at 90° for the Plantower, though from a physical inspection it does appear there is no real focusing of the collected light. The Shinyei would likely be classified as a photometer, which is considered a subset of the nephelometer in this work.

> We have computed results for two alternate optical setups, one with a collection angle of 90 +/- 45 degrees, and another with a collection angle of 45 +/- 5 degrees. These results are shown below. While the range for which a 1:1 value is obtained do change with collection angle (as expected), the results are qualitatively the same. A section on this has been added to the SI, along with a new Figure S4 (shown below); this is referred to in the main text as well (P22 lines 3-5).

[Figure]

*Finally, a key difference between the results presented here and sensor performance in the real world is that manufacturers may calibrate the sensor to ambient urban aerosol. This appears to be the case for Plantower - we were told (Malings et al. 2019) they calibrate to reference monitors in Chinese cities. (Met-One NPM is calibrated with 600 nm PSL.) Would it be possible to use a "typical" Beijing PSD instead of the ammonium sulfate calibration basis in the nephelometer results presented here?*

We chose polydisperse ammonium sulfate as a standard for comparison across all nephelometer types. A Plantower-specific study should definitely include Beijing urban aerosol as the calibrant. We are currently working on applying opcsim to the Plantower, but as noted above, focusing on specific sensors is beyond the scope of this work.

*I admit some of this may be more complicated and a lot more work, but people wanting to use the opcsim software may invariably want to do just that... And might turn to Dr Hagan for assistance anyway! Might as well get ahead of the curve and also get it published.*

> We agree and are working on specific use-cases; but we also didn't want to wait until that work was completed to describe (and release) the model. The number of specific sensors, aerosol types, and environmental conditions is too large to cover in a single study; so instead, we hope this can be useful to others to examine specific use cases. Each reader/user may want to simulate their sensor in a slightly different way with different calibrants and/or sensor specifications. To make this easier, extensive documentation showing exactly how to do this using this model can be found in the code's documentation (https://dhhagan.github.io/opcsim/). Questions pertaining to the code itself or to specific examples can be posted publicly on the GitHub repository itself (https://github.com/dhhagan/opcsim).

**Minor Comments**

*Page 15, line 8: what is "actual PM2.5 mass"? Is that the mass at 35% RH (like EPA regulations)? Please specify.*

> This paper calculates 'actual' mass at 0% RH. The difference between mass at 0% and 35% RH is negligible for most aerosols. Per Figure 2, with the exception of marine aerosol, kappa values are 0.25 or less. At a kappa of 0.25, the expected difference in mass at 35% relative to 0% RH is ~10%. This has been clarified in the text. (P 15, lines 8-9)

*Page 16, lines 18-20: wouldn't the "missing mass" problem be present at all humidities, just (over)compensated at higher RH due to hygroscopic growth?*

> Yes, the reviewer is correct. The missing mass problem is present at all humidities and is simply over-compensated at higher RH. The text has been updated (P 16 lines 17-19) to make this clearer: "In addition to overestimating mass loadings at high relative humidity due to hygroscopic growth, the OPCs underestimate the mass loadings across all relative humidities. This is not caused by relative humidity or a lack of hygroscopic growth, but instead is a result of the "missing mass" below the detectable threshold of the OPC."

*Figure 2: Some of these results are hard to decipher on a log scale. Maybe show it on a linear scale? (The marine case seems to be an extreme/can be moved to SI?)*

> We considered this, for the exact reasons the reviewer states. However, the linear scale actually makes the data harder to interpret, as the values for different aerosol types appear closer together. The linear-scale image (without marine) is shown below for reference.

Thus, we prefer to keep the figure with a log axis, to highlight differences between different particle types.

[Figure]

*Also in Fig 2, at least in Malings et al. (2019), we showed errors in the as-reported Plantower data as a function of RH (Table S2). Perhaps a more RH-resolved comparison could be made?*

This was the primary objective of Figure 2 (which is expressed as relative error rather than absolute error). However, these are calculations on model aerosol and not measurements of real-world particles (as in Malings 2020), so we prefer not to tabulate results, which might give the wrong impression about the applicability of these errors to real-word systems/sensors.

*Page 18, lines 4-6: Can RI changes really cause such significant mis-assignment? Maybe at the margins/bin boundaries, sure. But the PSL, SOA, ammonium sulfate Mie curves (Fig 3) are pretty close to each other, and these are the dominant PM2.5 components by mass and (probably, OPC-wise anyway) number.*

This is an important point, which we did not address explicitly in the original submission. We have added an additional section and multiple figures to the SI that shows just how much of an impact slight changes in RI can have on the bin assignment for an OPC. This new section in the SI shows that yes, even small changes in RI can lead to large changes in how a particle is assigned to a bin – it can span many bins for a given particle size and depends strongly on the properties of the OPC as well as the aerosol itself. These text and figures are given below:

**Impact of changes in aerosol optical properties on OPC bin assignment**

Here we investigate how bin assignment may be affected If the aerosol optical properties of the measured aerosol are different than the material used to calibrate an OPC. We compute the light scattering for a range of possible refractive index values and plot them

in Figure S2 below for a 600 nm particle. Across a wide range of optical properties, the particle can be placed in 3 different bins or left out completely ("NO BIN"). However, it should be noted that fairly large changes in either the real or imaginary part of the RI need to change in order for a particle to be assigned to a different bin.

[Figure]

**Figure S2**. Bin assignment as a function of refractive index for a 600 nm particle. An OPC with 16 bins between 380 nm and 17.5 µm and a 658 nm laser calibrated using PSLs was used to generate this data. Data are colored by the computed integrated scattering cross section.

If we complete the same exercise for a 1 µm particle, the results are quite different. Small changes in the absorbing component of the refractive index can lead to large differences in how the particle is assigned to a bin. Over the entire range of optical properties shown in Figure S3, the 1µm particle can be assigned to 5 different bins (or NO BIN). For this specific OPC, this means the 1 µm particle can be assigned a diameter of anywhere from 0 µm to 1.26 µm. This indicates that for large particles, differences in optical properties of calibration aerosol and measured aerosol can lead to large errors in bin assignments (and hence mass measurements).

[Figure]

**Figure S3.** Bin assignment as a function of refractive index for a 600 nm particle. An OPC with 16 bins between 380 nm and 17.5 µm and a 658 nm laser calibrated using PSLs was used to generate this data. Data are colored by the computed integrated scattering cross section.

*Page 19, line 9: "calibrated using PSL", not ammonium sulfate? (PSL matches the RI in parentheses and the caption of Fig 4.)*

Thank you for pointing out this typo – this has been updated in the text.

*Pages 24-25: "some proxy for aerosol composition" - In Malings et al. (2019), we used PM composition data from the EPA CSN network (139 sites across the US, Fig S4) with a wide range of chemical composition and found the results did not change significantly (Fig S5), so long as some fRH correction was used. Might be relevant here*

This is a very good point made by the reviewer, but that data is not available in real-time, thereby limiting its utility in the context of real-time, low-cost sensor networks. If post-processed data is desired, there are certainly ways to correct the data from low-cost particle sensors for compositional effects. However, this is not a viable approach for realtime data. To clarify we are referring to real-time corrections, we have changed the text in _ to read: "…it would be possible to vastly reduce this error and improve the accuracy of mass measurements using OPSs for real-time data collection."

*Page 25, line 11: "therefore" not "therefor"*

This has been corrected.

*Page 25, line 24: are urban OECD aerosol size distributions "highly variable"? This might be another example of "errors that may not be significant in the US/EU but are important in developing countries".*

While the mean aerosol size distribution in OECD countries may be somewhat normalized over time, it is certainly variable on a minute-by-minute (or faster) time basis, which is often the focus of low-cost, real-time air quality sensor networks. To clarify that we were referring to general variability, we have removed the word 'highly' from the text.

*Table 4: some issues with this - "small PSD" or "large PSD" suggests narrow or broad size distribution, but I suspect the authors mean "smaller aerosols" or "larger aerosols". Please clarify.*

We thank the reviewer for pointing this out – we have clarified this in the text by changing "PSD" to "GMD", indicating that we are talking about smaller particle sizes.

*NIST urban aerosol - SRM 1648 was collected in 1976-1977 and may not be representative of, well, anything these days...*

This is an excellent point! We are aware of no other standards for urban aerosol, so we have added "or collected particles" to this entry.

*Maybe the "aerosol properties" column should have some references?*

This column has been updated to include references.

*The authors emphasize this table is only a start, so it's fine to include it, I guess. There could be an entire workshop devoted to Table 4...*

We are in complete agreement. We thank the author for pointing this out and agree that much more could be investigated here; however, as noted above, this is not the focus of this paper and we hope that this will be expanded in the future.

*Throughout: Malings et al. (2019), not Malings et al. (2020). (I don't know why T&F's "download citation" shows the year as 2020. The paper was published in June 2019 and my downloaded PDF shows 2019 as the year for citation.)*

> The AS&T special issue on low-cost sensors (Vol. 54 issue 2) was published in 2020, despite the fact that most of the papers were posted online in 2019. Since 2020 is the official publication date, we leave the reference as-is.

*Throughout: "The pluralization of abbreviations, too, requires no apostrophes. More than one CD = CDs... Etc." - Benjamin Dreyer, Dreyer's English (2019). See also: https://www.chicagomanualofstyle.org/qanda/data/faq/topics/Plurals.html?page=1*

> Thank you for pointing this out – this has been fixed throughout the text.

**Reviewer 3**

**Major Comments**

*More analysis needs to be done to demonstrate the importance of these factors in measuring real world particles. Examples presented in the text are simple hypothetical cases that just show the importance of each factor. It is important for the reader to see how much the results of actual atmospheric particle measurements get affected by these errors. Analysis of data from previous studies would be helpful to obtain a picture of how much those data would change if corrected for the effects. One major question is to see if the final conclusion from some of previous studies would be affected based on these factors.*

> We thank the reviewer for their suggestions (similar to those made by the other two reviewers) and absolutely agree that real-world examples and data analysis would be of interest and importance. We also agree that these examples are purely hypothetical and show the importance of each factor. But this in fact was our objective – our goal was to show the impact of each individual factor via modeling (as opposed to field measurements) for the reasons outlined in the introduction of this paper. Including more real-world data is beyond the scope of this work, but we do plan on follow-up work that compares our modeled results to field measurements for some nephelometers and OPCs and hope others will carry out similar studies for individual use-cases.

*The abstract and introduction create the expectation that this article would also address the interaction between different factors and the impact on accuracy of measurement results. However, presented cases are all single-factor studies. If authors believe the interdependence of these factors is significant, these mutual effects should be addressed in the paper.*

> While we examine the different factors individually, many of these interdependencies are actually built in: for example, water uptake affects size as well as density and RI (p. 5, lines

8-11). We did attempt to examine all of these factors in aggregate by varying all of them across atmospherically relevant ranges; but such reasonable ranges can vary dramatically from environment to environment, and the errors were largely a function of the spread in values chosen, and not anything inherent about the sensors themselves. To avoid such ambiguities in ranges of aerosol types, we instead present broad uncertainty ranges from each contributor in Table 3 to give the reader an overall sense of how the different sources of error contribute. To avoid implying in the abstract and introduction that these interdependencies are systematically examined, we have clarified the text in a few places:

P2 line 13: "…is used to estimate the fractional error in mass loading…"
P5 line 8: "…to our knowledge, there has not been a systematic, comprehensive investigation of all these factors together."

**Minor Comments**

*Title – "low-cost" word can be omitted from the title as this manuscript looks at low cast as well as "high end" instruments.*

Because the focus and objective of this work is investigating the accuracy of low-cost sensors, we feel it is important to keep the term in the title, even if the work is applicable to higher-fidelity instruments as well.

*P5, L3 – Please provide a brief description of factors resulting in "changes in aerosol optical properties".*

We have changed the text to make this more clear. The text now reads: "…(3) changes in scattering efficiency due to differences in aerosol optical properties; and (4) the need for aerosol-specific correction factors to account for differences in density."

*P8, Table 1 – Have the authors tried contacting the manufacturers to obtain the data that is missing in the literature?*

We had been relying on what was in the literature, but on the reviewer's suggestion, have reached out to the manufacturers. We have not yet received responses but will update the table when/if we do.

*P12, L7 – Why is geometric mean used for mass calculations? Are the authors suggesting that they use the measured pulse heights from individual particles rather than assuming uniform size distribution across each individual size bin? If yes, what is the point of referring to size bins?*

The geometric mean is used for mass calculations because when using an OPC, the exact particle diameters are unknown. Instead, only the number of particles in a given size bin is known. To model the process of an OPC as closely to reality as possible, we take the exact size of a particle, compute its scattering value, and then assign that to a bin, which is the process an actual OPC takes when making measurements. We bin the resulting scattering values in the same way an actual OPC would (an analog-to-digital converter or FPGA would bin them in this context) which is important for understanding small changes in aerosol properties can lead to a particle being sized incorrectly and placed into the wrong bin. Thus, it is quite important that we refer to bins as opposed to the actual size of the particle, as this is how the devices work in reality.

*P12, L24 – Similar comment as above. Please clarify the reason for preferring geometric mean diameter.*

Data from an OPC does not include exact particle diameters and is the exact light pulse height is not typically stored. If pulse heights were stored, then theoretically, one could correlate the pulse height to a particle diameter; however, this is not how OPCs typically work. Whether using a low-cost OPC or a reference-grade OPC data is in the form of a histogram where the bins represent a range of particle diameters and the height represents the number of particles or particle number concentration in that size bin. To then compute a mass value, it is necessary to assume some diameter or volume per particle. As mentioned in the paper, this is typically the geometric mean diameter.

*P15, L9-10 –"overestimate" and "underestimate" would be more accurate terms for these sentences.*

This has been corrected to reflect these terms.

*P15, L22-24 – The wording in this section is unclear. Do the authors mean that "very few low cost OPSs control relative humidity. . ."?*

The reviewer was correct – this has been corrected.

*P16, L1- Should read "Figure 2 shows the impact that RH can. . ."*

This has been corrected.

*P17, L20 – Please add more explanation about Mie scattering of black carbon particles in the <300nm range. The trend shown in >300nm range is understandable based on the BC particles being more absorbing, but an explanation for <300nm range is missing.*

BC particles smaller than 300 nm scatter more light than non-BC particles at this size because the scattering component of the refractive index is much higher; for black

carbon, the refractive index is 1.95 – 0.79j. When the particles are smaller, this has a greater effect than the absorbing component, as is discussed in Bohren and Huffman. This effect can also be seen in Figure 3 – the Mie calculations show that BC < 300 nm scatters more light than non-BC particles, but as the absorbing component becomes more relevant as the particles get larger, BC scatters far less light than the non-BC particles.

*Fig 5 – Please show the contour line corresponding to Mm/Ma=1*

This suggestion has been implemented – there is now a black dashed line indicating the 1:1 line on Figure 5. The image is also re-printed below.

[Figure]

*Fig 5 – It seems that even at the calibration conditions (GM=400 nm, and GSD=1.65), mass loading accuracy is not equal to unity in any of the plots. What is the explanation?*

The OPCs will almost always be unable to see a portion of the size distribution, even if it is very large, as the tail of the distribution will fall below the detectable range of the OPC. This 'missing mass' is discussed throughout the section. As a larger fraction of the size

distribution is within the detectable range of the OPC, the more accurate the device will become. As for the nephelometer, an incorrect refractive index (1.592) was used instead of the listed (ammonium sulfate, 1.521). The figure has been updated. At this distribution, the value is 1 at these conditions and changes as the distribution changes, as is discussed in the section itself.

*Table 4 – Need to mention that the main metric in consideration in this table is mass loading. Clearly, if the focus in a study is on evolution of particle size distribution rather than integral mass loading, nephelometer can't be recommended.*

We have updated the description of the table to make it clear the primary metric is accuracy in mass loading measurement.